# Small molecule inhibition of ATM kinase increases CRISPR-Cas9 1-bp insertion frequency

Heysol C. Bermudez-Cabrera[1], Sannie Culbertson[1], Sammy Barkal[1], Benjamin Holmes[2,3,4], Max W. Shen [5], Sophia Zhang[1], David K. Gifford [2,3,4] & Richard I. Sherwood [1,6 ✉]

Mutational outcomes following CRISPR-Cas9-nuclease cutting in mammalian cells have recently been shown to be predictable and, in certain cases, skewed toward single genotypes. However, the ability to control these outcomes remains limited, especially for 1-bp insertions, a common and therapeutically relevant class of repair outcomes. Here, through a small molecule screen, we identify the ATM kinase inhibitor KU-60019 as a compound capable of reproducibly increasing the fraction of 1-bp insertions relative to other Cas9 repair outcomes. Small molecule or genetic ATM inhibition increases 1-bp insertion outcome fraction across three human and mouse cell lines, two Cas9 species, and dozens of target sites, although concomitantly reducing the fraction of edited alleles. Notably, KU-60019 increases the relative frequency of 1-bp insertions to over 80% of edited alleles at several native human genomic loci and improves the efficiency of correction for pathogenic 1-bp deletion variants. The ability to increase 1-bp insertion frequency adds another dimension to precise template-free Cas9-nuclease genome editing.

[1] Division of Genetics, Department of Medicine, Brigham and Women's Hospital and Harvard Medical School, Boston, MA, USA. [2] Computer Science and Artificial Intelligence Laboratory, Massachusetts Institute of Technology, Cambridge, MA, USA. [3] Department of Biological Engineering, Massachusetts Institute of Technology, Cambridge, MA, USA. [4] Department of Electrical Engineering and Computer Science, Massachusetts Institute of Technology, Cambridge, MA, USA. [5] Computational and Systems Biology Program, Massachusetts Institute of Technology, Cambridge, MA, USA. [6] Hubrecht Institute, 3584 Utrecht, CT, The Netherlands. ✉email: rsherwood@bwh.harvard.edu

The CRISPR-Cas9 system is a powerful and versatile tool for precision genome editing, and it has shown potential as a therapeutic tool for treating genetic diseases. Originally developed from a bacterial immune system, the CRISPR system was modified for experimental use in mammalian cells by engineering single guide RNAs (gRNAs) that complex with a CRISPR-associated nuclease protein, most commonly Cas9, to induce double-stranded breaks (DSBs) at a specific location in the genome[1,2]. Unlike in bacteria where cleaved DNA is degraded, when a mammalian genome undergoes a DSB, the damage is repaired by endogenous cellular machinery.

Numerous DNA repair pathways are employed to repair Cas9-induced DSBs. Three of the most prominent pathways are the non- homologous end joining (NHEJ) pathway, the microhomology-mediated end joining (MMEJ) pathway, and the homologous recombination (HR) pathway[3,4]. Most data on the repair of Cas9-induced DSBs comes from retrospective analysis of mutant genotypes, and thus it is largely unknown how frequently DSBs are repaired perfectly only to be re-cleaved. NHEJ starts when blunt, broken ends are recognized by Ku, a eukaryotic dimeric protein complex composed of polypeptides Ku70 and Ku80[5]. Ku then complexes with DNA-PKcs, A DNA-dependent protein kinase whose autophosphorylation recruits and directs DNA ligase IV, XLF, and other accessory factors to the DSB[6]. This complex, called the paired-end complex, ligates the broken DNA ends back together. NHEJ tends to be the fastest method of repairing Cas9-induced DSBs and can complete a DSB repair as soon as 30 min after the cut[7]. Although NHEJ is generally reliable and error-free, if the broken DNA ends are not perfectly blunted, small insertion or deletion mutations are likely to occur[8].

The MMEJ pathway involves resection of the 5′ ends on both sides of the cleavage site, which allows micro-homologous sequences flanking the cutsite to align through base pairing, commonly causing deletion mutations[9]. Although much of the MMEJ pathway is still unknown, the resection of 5′ ends has been linked to Replication Protein A, and the entire MMEJ pathway is independent of many proteins linked to the NHEJ pathway[10]. HR is a DNA repair mechanism with high fidelity that relies on an intact DNA template that is identical to the DSB region[11]. Such a template would only be expected to exist natively during certain cell cycle phases (G2 and mitosis) as a sister chromatid or with the introduction of a synthetic homology-directed repair template; as such, it is not thought to contribute substantially to mutagenic outcomes of "template-free" Cas9 editing experiments.

Recently, several studies have shown that the mutational outcomes of CRISPR-Cas9 are predictable based on the genomic sequence surrounding the cutsite[12–14]. By analyzing mutational outcome data at thousands of Cas9 DSBs, these studies found that the vast majority of all mutations could be classified as microhomology-containing deletions (MH deletions), non-microhomology deletions (non-MH deletions), and 1-bp insertions. MH deletions are defined as deletions that could have arisen through micro-homologous base-pairing using the MMEJ mechanism, and their frequency depends on the length and GC content of microhomology. In certain cell lines, MH deletions are the dominant mutagenic outcome, suggesting the robust involvement of MMEJ in the repair of Cas9 mutations[12]. Non-MH deletions are deletions that could not have arisen through microhomology; their frequency decreases exponentially with increasing deletion length, and they may be attributable to NHEJ. The relative frequency of 1-bp insertion outcomes depends on the identity of the nucleotides flanking the Cas9 cutsite, most frequently resulting in duplication of the nucleotide flanking the cutsite distal to the PAM. 1-bp insertions are thought to arise from NHEJ repair of staggered Cas9 DSBs filled in by gap-filling polymerases[15].

Using these data and observations, groups have developed machine-learning algorithms, such as inDelphi that accurately predict the genotypes and frequencies of Cas9-induced mutations for a given genomic target sequence[12]. Interestingly, inDelphi predicts that Cas9 editing at 5–11% of genome-targeting gRNAs, depending on the cell type used, should result in a single dominant mutation occurring in over 50% of all mutated genomes. These "Precision50" gRNAs have been empirically validated to result in precise editing, with some Precision50 gRNAs yielding a predominant MH deletion product and others giving a predominant 1-bp insertion product. Precision50 gRNAs represent an intriguing set of sites for disease modeling and therapeutic editing. The ability to predict the mutational outcomes of CRISPR-Cas9 edits and, in certain cases, to achieve a single dominant mutant product introduces the possibility of precisely creating or fixing pathogenic mutations. In fact, such precise generation and correction of pathogenic alleles through template-free Cas9-nuclease editing has been demonstrated[16–18].

However, even in the most precise cases, Cas9-nuclease repair only yields the dominant product 50–80% of the time. Thus, in order to create or fix specific mutations, there is a clear need for approaches that increase the chance of a certain mutational outcome. It has been shown that the DNA-PK inhibitors NU7026 and DNA-dependent protein kinase inhibitor III increase the frequency of Cas9-mediated MH deletions in mouse embryonic stem cell (mESC)[12] and that introducing microhomology sites near the Cas9 cutsite can increase MH deletions[19], but approaches to increase 1-bp insertion outcomes are lacking.

Here, we use a fluorescent assay for MH deletion repair outcome frequency to screen 487 small molecules for their effects on CRISPR-Cas9 mutational outcomes. Through such screening, we identify small molecules capable of increasing MH deletion outcomes as well as 1-bp insertion outcomes. Notably, we show that an Ataxia Telangiectasia Mutated (ATM) kinase inhibitor, KU-60019, significantly increases Cas9 1-bp insertion frequency in multiple cell lines and for multiple Cas9 species. This effect is replicable in cells with *Atm* knockout and with three distinct ATM small molecule inhibitors, indicating a role for ATM in inhibiting 1-bp insertion outcomes. KU-60019 is capable of further increasing the precision of 1-bp insertion outcomes of Precision50 gRNAs, confirming a molecule capable of altering template-free CRISPR-Cas9 repair outcomes.

## Results

**Small molecule library screening for altered Cas9 repair outcomes.** In order to identify small molecules that alter template-free Cas9 repair outcomes, we used a previously published reporter construct in which MH deletion outcomes can be monitored through flow cytometry. The construct encodes a constitutively expressed LDL receptor (LDLR) gene with a frameshift microduplication followed by a 2A peptide and GFP (LDLR-Dup, Supplementary Fig. 1, Supplementary Data File 1)[1]. We stably integrated a single copy of this LDLR-Dup construct into mESCs through Tol2 transposition. In the absence of Cas9 editing, cells do not express GFP and uptake minimal LDL; however, upon treatment with *Streptococcus Pyogenes* Cas9 (SpCas9) and a gRNA targeting the microduplication, a large fraction of cells become GFP+, increase LDL uptake, and show genotypic repair to wild-type LDLR genotype through microhomology deletions (MH deletions)[1]. We reasoned that small molecules that alter the ratio of MH repair outcomes vs. other repair outcome classes would show altered fractions of GFP+ LDL+ cells.

We screened a set of 487 small molecules annotated to impact DNA repair (Supplementary Data File 2). LDLR-Dup mESCs

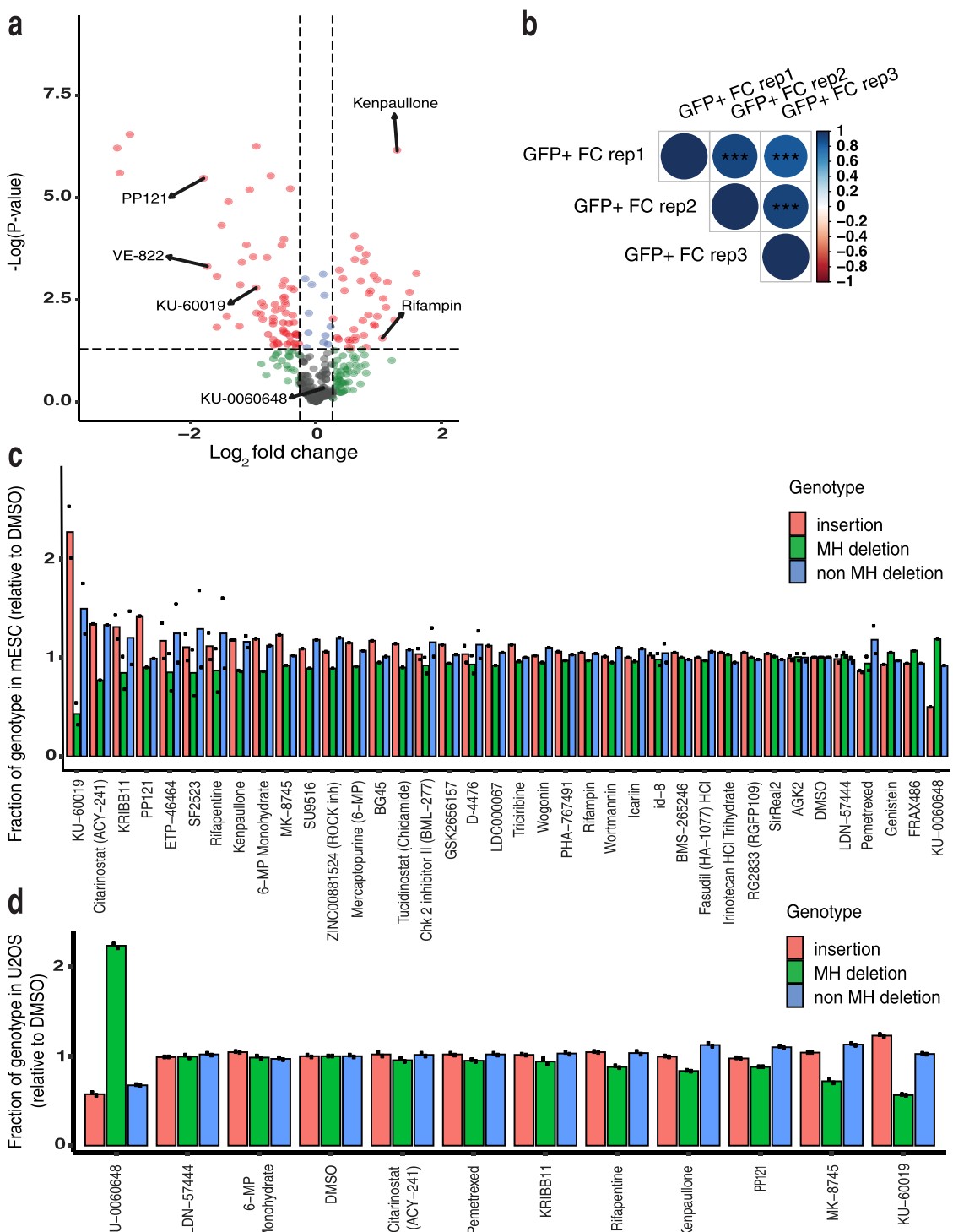

**Fig. 1 Identification of small molecules that alter SpCas9 mutational outcomes. a** Volcano plot exhibiting the log2 fold changes (FC) in GFP+ cells relative to the DMSO control for the 318 small molecules at 2 μM. Small molecules yielding a (less than or greater than) log2 FC of 0.26 of normalized GFP activity and a *p* value <0.05 were identified as hits and are exhibited as red dots. *P* values calculated using two-sided Welch's t-test. **b** Replicate consistency of FC GFP+ population for assay at 2 μM concentration (*N* = 3). FC was normalized by the GFP signal from DMSO-treated wells. Spearman correlation, *p* value < 0.0001. Average fraction of each genotype among edited reads relative to DMSO observed in mESC (**c**) or U2OS (**d**) transfected with the 48-site library (*N* = 1 or *N* = 2 biologically independent samples, as shown). ***p < 0.001.

were transfected with a single construct containing Cas9, a LDLR-Dup-targeting gRNA, and a Neomycin resistance cassette along with small molecules dosed at 2 μM (Fig. 1a, b, Supplementary Figs. 1 and 2a, b) or 8 μM (Supplementary Fig. 2c–f) in three replicates each. Transient neomycin selection was performed post

transfection to select for cells that received Cas9 and gRNA, and small molecule treatment was given for the first 72 h after transfection. Seven to eight days after transfection, cells were incubated with fluorescently tagged LDL cholesterol, and flow cytometry was performed to measure GFP expression and LDL

uptake for all wells, including DMSO-treated controls (Supplementary Fig. 3). Of the 487 small molecules, 318 and 294 molecules had sufficient cell survival to obtain valid data at 2 and 8 µM, respectively (Fig. 1a, b, Supplementary Fig. 2, Supplementary Data File 3). Replicate consistency was relatively high across replicates (Spearman $r = 0.48$–0.52, Fig. 1b). We chose to focus on small molecules that significantly changed the fraction of GFP + cells, as LDL uptake was variable among replicates. At the 2 µM dose, we observed 38 small molecules that significantly increased the fraction of GFP+ cells and 63 small molecules that significantly decreased the fraction of GFP+ cells compared to the DMSO control ($p < 0.05$, Fig. 1a, b). We performed an additional set of screening on 61 small molecules from the initial screen at 2 and 8 µM, purchased from a different supplier to ensure the effects observed were molecule-specific. We found that many molecules reproducibly affected the GFP+ fraction across several screens. Among those molecules increasing the GFP+ fraction were DNA/RNA polymerase inhibitors, such as Rifapentin and CDK inhibitors, such as Kenpaullone. Among the small molecules that consistently decreased GFP+ fraction DNA-PK inhibitor (PP121), and ATM/ATR inhibitors, such as KU-60019 and VE-822 (Fig. 1a, Supplementary Fig. 2).

To determine in more detail how small molecules from our screen influence Cas9 repair outcomes, we followed up on 38 compounds (Supplementary Data File 4) which showed a significant change in the fraction of LDLR-Dup GFP+ cells within the population, suggesting possible alteration of Cas9 repair outcome distribution. We utilized a library consisting of 48 gRNA:target pairs encompassing 16 precise MH deletion sites, 16 precise 1-bp insertion sites, and 16 sites with average outcome distribution derived from previously published 2000-member libraries[12,20]. We refer to this 48 gRNA:target library as the '48-site library' (Fig. 1c, d, Supplementary Data File 5). We note that library approaches, such as the 48-site library only measure cutsite-proximal insertions and deletions that do not disrupt the flanking PCR primers, so our data cannot measure the frequency of long deletions that can be induced by Cas9-nuclease editing[21]. mESC and human U2OS osteosarcoma cells stably expressing the 48-site library were transfected with Cas9 and treated with one of 38 significant small molecules from the initial screen, each in two biological replicates.

We identified two small molecules that increased either 1-bp insertions (KU-60019) or MH deletions (KU-0060648) in both mESC and U2OS (Fig. 1c, d, Supplementary Data File 6). On the whole, U2OS cells favor insertion outcomes and mESCs favor MH deletion outcomes in the absence of small molecules, so it is not surprising that we found KU-60019 has larger effects in mESCs where insertions are normally low and KU-0060648 has larger effects in U2OS where MH deletions are low. Several other compounds had less robust impacts on outcome distribution, while a larger number affected the Cas9 editing rate without significantly altering the Cas9 repair outcome distribution (Supplementary Fig. 2c, d). The primary screen did not provide a way to distinguish altered MH deletion from altered overall editing rate, and these 48-site library results show that a majority of hits from the primary screen have a larger effect on editing rate than outcome distribution. For this reason, we chose to follow up exclusively on KU-60019 and KU-0060648 for the remainder of the study.

**KU-60019 and KU-0060648 change mutational outcome distribution and editing frequency in a dose-dependent manner.** KU-60019, an inhibitor of ATM kinase, has been shown to influence cellular response to DNA damage and cell cycle checkpoint activation, but little is known about the effect of ATM

or this small molecule inhibitor on DSB repair outcomes[3,4]. KU-0060648, is an inhibitor of DNA-PK, a key enzyme in the NHEJ repair pathway. Other DNA-PK inhibitors have been used to alter Cas9 outcomes, promoting MH deletions, and HDR outcomes[22,23]. To understand the relationship between concentration and activity for these compounds, we performed a titration of KU-60019, KU-0060648, and the previously reported DNA-PK inhibitor NU7026 in 48-site library-containing mESC cells. We confirmed that KU-60019 increased 1-bp insertions (fold change: 1.2–1.6 for 2 and 4 µM) while decreasing MH deletions (fold change: 0.91–0.74) in a dose-dependent manner (Fig. 2a). KU-0060648 increased the MH deletion fraction in a dose-dependent manner (fold change: 1.03–1.07) at the expense of 1-bp insertions (fold change: 0.82-0.5 for all concentrations, Fig. 2b). The increase in MH deletions induced by KU-0060648 was of similar magnitude to that induced by NU7026 (Fig. 2c), suggesting that both compounds provide similarly effective NHEJ inhibition. All of these compounds also dose-dependently decreased the fraction of Cas9-edited alleles (fold change of edited allele fraction with increasing dosing: 0.77–0.33, 0.69–0.29, 0.84–0.56 for KU-60019, KU-0060648, NU7026, respectively Fig. 2d–f) with a decrease for KU-60019 at 2 and 4 µM (Fig. 2d) and KU-0060648 at 1 and 2 µM (Fig. 2e). We did not observe significant changes in induction of cell death by either KU-60019 and KU-0060648 compared to DMSO at the tested doses (Supplementary Fig. 4a).

Notably, the 48-site library includes eight known pathogenic 1-bp deletion alleles causal to diseases, such as familial hypercholesterolemia (LDLR) and neurofibromatosis (NF2), for which 1-bp insertion editing preferentially repairs the allele to wild-type genotype. KU-60019 at 2 µM increased the relative frequency of 1-bp insertion pathogenic allele repair in seven of eight of these loci, by on average 1.49-fold relative to DMSO with a maximum of 1.99-fold 1-bp insertion (Supplementary Fig. 5a).

We observed a similar increase in the relative frequency of 1-bp insertions (fold change: 1.32, $p < 0.001$, Supplementary Fig. 6a) and concomitant decrease in total editing efficiency (fold change: 0.38, Supplementary Fig. 6b) in the presence of KU-60019 as compared to DMSO when Cas9 and gRNA were delivered via ribonucleoprotein (RNP), suggesting that these effects are independent of changes in transgene expression after plasmid transfection.

**KU-60019 increases 1-bp insertions through an ATM inhibition mediated pathway.** In order to address whether KU-60019 increases 1-bp insertion frequency through ATM inhibition and not an alternative pathway, we investigated the effects of *Atm* knockout on Cas9 mutational outcomes. mESCs expressing a GFP transgene were dosed with Cas9 and a gRNA targeting *Atm* or a non-targeting control. The cells were subsequently transfected with three different gRNAs targeting GFP. All 3 GFP-targeting gRNAs exhibited a robust increase in the fraction of 1-bp insertions among all edited reads in the *Atm* knockout mESCs as compared to control mESCs (fold change: 1.27–1.53, Fig. 3a). In addition, the total editing efficiency was significantly lower in *Atm* knockout mESCs as compared to control mESCs for all three gRNAs (fold change: 0.17–0.35, Fig. 3b), suggesting that *Atm* presence promotes mutagenic outcomes of Cas9 editing. Furthermore, we investigated the effects of three additional ATM inhibitors on mutational outcomes in the 48-site library; CP-466722, AZD1390, and AZ32[24–26]. All three inhibitors increased the fraction of 1-bp insertions (fold change: 1.43–1.82, Fig. 3c), suggesting that KU-60019 increases the frequency of 1-bp insertions through its inhibition of ATM. Notably, we also observed that AZD1390, the ATM inhibitor with the greatest increase in 1-bp insertion frequency (fold change: 1.82, $p < 0.001$), also induced the greatest decrease in overall editing efficiency (fold

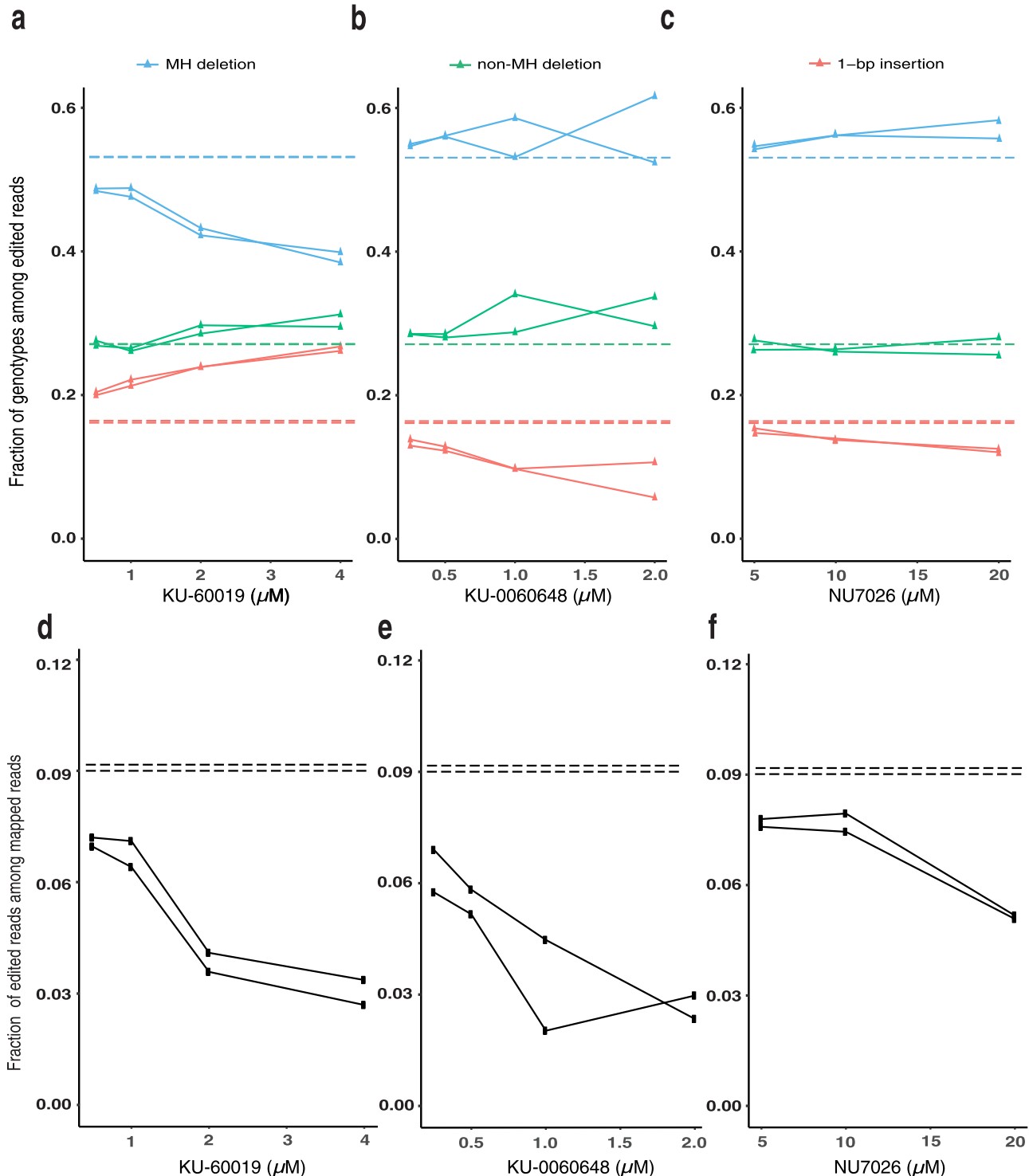

**Fig. 2 KU-60019 induces dose-dependent increases in 1-bp insertion outcomes in mESC.** Fraction of 1-bp insertion, MH deletion, and non-MH deletions among all SpCas9-NG-edited reads at different µM concentrations for **a** KU-60019, **b** KU-0060648, or **c** NU7026. Fraction of SpCas9-NG-edited reads among the total mapped reads for mESC with 48-site library treated with DMSO (horizontal line), KU-60019 (**d**), KU-0060648 (**e**), or NU7026 (**f**) at different µM concentrations as noted. Each plot shows the 48-site library averaged genotype fraction, $N = 2$ biologically independent samples. The horizontal dashed lines mark the DMSO genotype fraction for either replicate, respectively. The solid lines mark the genotype fraction for each experimental replicate, respectively and closely overlap.

change: 0.36, $p < 0.001$, Fig. 3d). Taken together, these experiments indicate that reduction in ATM kinase activity increases the relative frequency of Cas9-mediated 1-bp insertion outcomes.

**KU-60019 increases 1-bp insertion frequency with a second Cas9 species and at native human loci.** To compare the effects of KU-60019 for different Cas9 species, we used a gRNA:target library

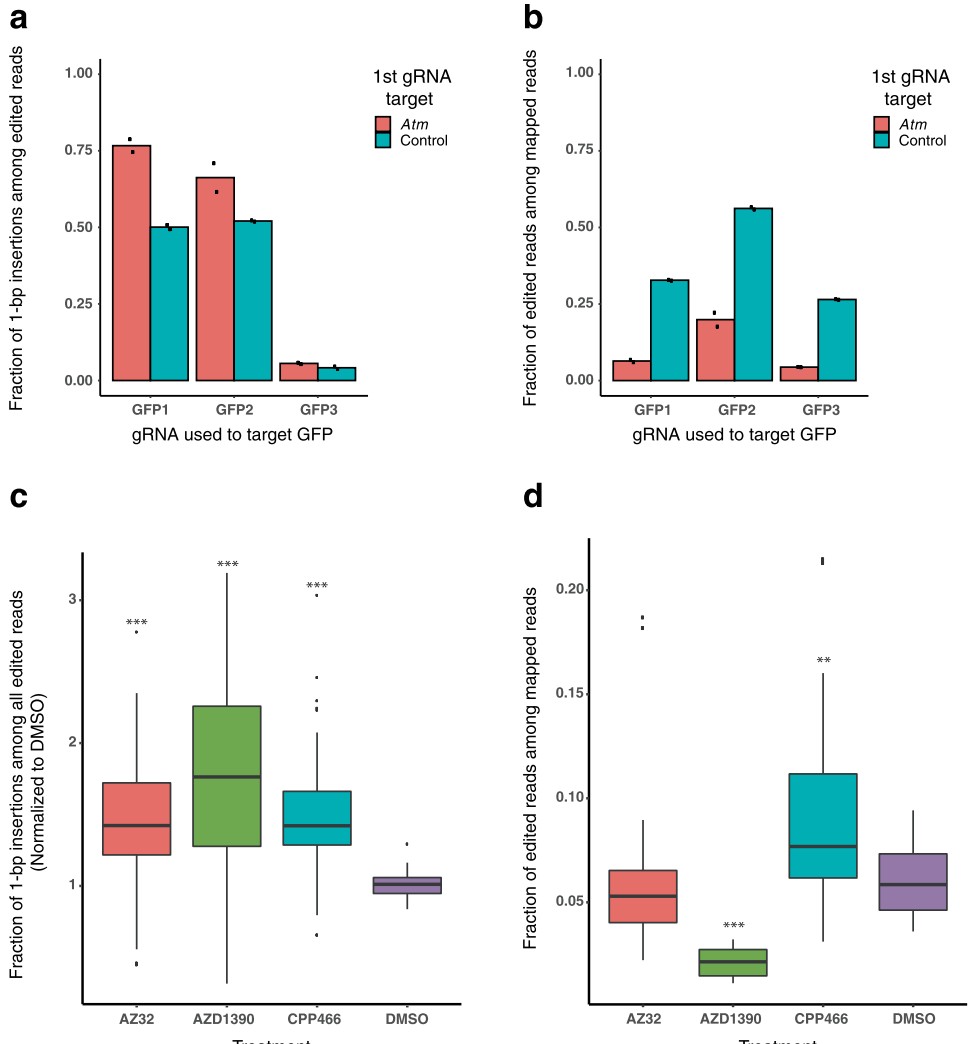

**Fig. 3 ATM loss-of-function through genetic knockout or small molecule inhibition leads to increased 1-bp insertion frequency. a** Bar chart showing the fraction of 1bp-insertion mutations among edited reads at three GFP-targeting gRNA sites in *Atm* or control knockout mESCs (*N* = 2 biologically independent samples). **b** Bar chart exhibiting the edited fraction of all mapped reads for each of the conditions described in **a**, respectively. **c** Boxplot exhibiting the fraction of 1-bp insertions observed in mESCs for each site of the 48-site library in the presence of known ATM inhibitors or DMSO (*N* = 3 biologically independent samples). **d** Boxplot exhibiting the editing frequency observed for each site depicted in **c**, respectively. *P* values were calculated using a one-way ANOVA with Tukey HSD).**$p < 0.01$, and ***$p < 0.001$. Boxplot statistics for 1-bp insertion fraction with AZ32: minima = 0.401, lower whisker bound = 0.447, lower box bound = 1.210, center = 1.423, upper box bound = 1.722, and upper whisker bound = 2.350. Boxplot statistics for 1-bp insertion fraction with AZD1390: minima = 0.316, lower whisker bound = 0.316, lower box bound = 1.268, center = 1.763, upper box bound = 2.261, upper whisker bound = 3.191, maxima = 3.191. Boxplot statistics for 1-bp insertion fraction with CPP466: minima = 0.743, lower whisker bound = 0.795, lower box bound = 1.286, center = 1.421, upper box bound = 1.662, and upper whisker bound = 2.074. Boxplot statistics for 1-bp insertion fraction with DMSO: minima = 0.837, lower whisker bound = 0.837, lower box bound = 0.947, center = 1.011, upper box bound = 1.060, and upper whisker bound = 1.163. Boxplot statistics for total edited fraction with AZ32: minima = 0.022, lower whisker bound = 0.022, lower box bound = 0.040, center = 0.053, upper box bound = 0.066, and upper whisker bound = 0.090. Boxplot statistics for total edited fraction with AZD1390: minima = 0.011, lower whisker bound = 0.011, lower box bound = 0.015, center = 0.021, upper box bound = 0.027, upper whisker bound = 0.032, and maxima = 0.032. Boxplot statistics for total edited fraction with CPP466: minima = 0.031, lower whisker bound = 0.031, lower box bound = 0.061, center = 0.077, upper box bound = 0.113, and upper whisker bound = 0.160. Boxplot statistics for total edited fraction with DMSO: minima = 0.036, lower whisker bound = 0.036, lower box bound = 0.046, center = 0.059, upper box bound = 0.075, upper whisker bound = 0.094, and maxima = 0.094.

with 12 pathogenic 1-bp deletion alleles causal to diseases, such as cystic fibrosis - a monogenic disease associated with the cystic fibrosis transmembrane conductance regulator (CFTR) gene - and familial adenomatous polyposis sites, for which 1-bp insertion editing with KKH-SaCas9, a smaller Cas9 orthologue shown to be amenable to in vivo gene editing[27,28], should preferentially restore the pathogenic allele to wild-type genotype. For all 11 gRNA:target pairs in which we detected successful editing, KKH-SaCas9

treatment of library-containing mESC cells in the presence of KU-60019 resulted in a significant 1.65 fold increase in the average 1-bp-insertion frequency (Supplementary Fig. 7a) at the expense of MH deletions and non-MH deletions. All 11 pathogenic alleles showed increased 1-bp insertion-mediated pathogenic allele repair with a maximum of 3.08-fold as much 1-bp insertion (Supplementary Fig. 5b). We note that, without the benefit of a predictive model of features influencing 1-bp insertion frequency for KKH-

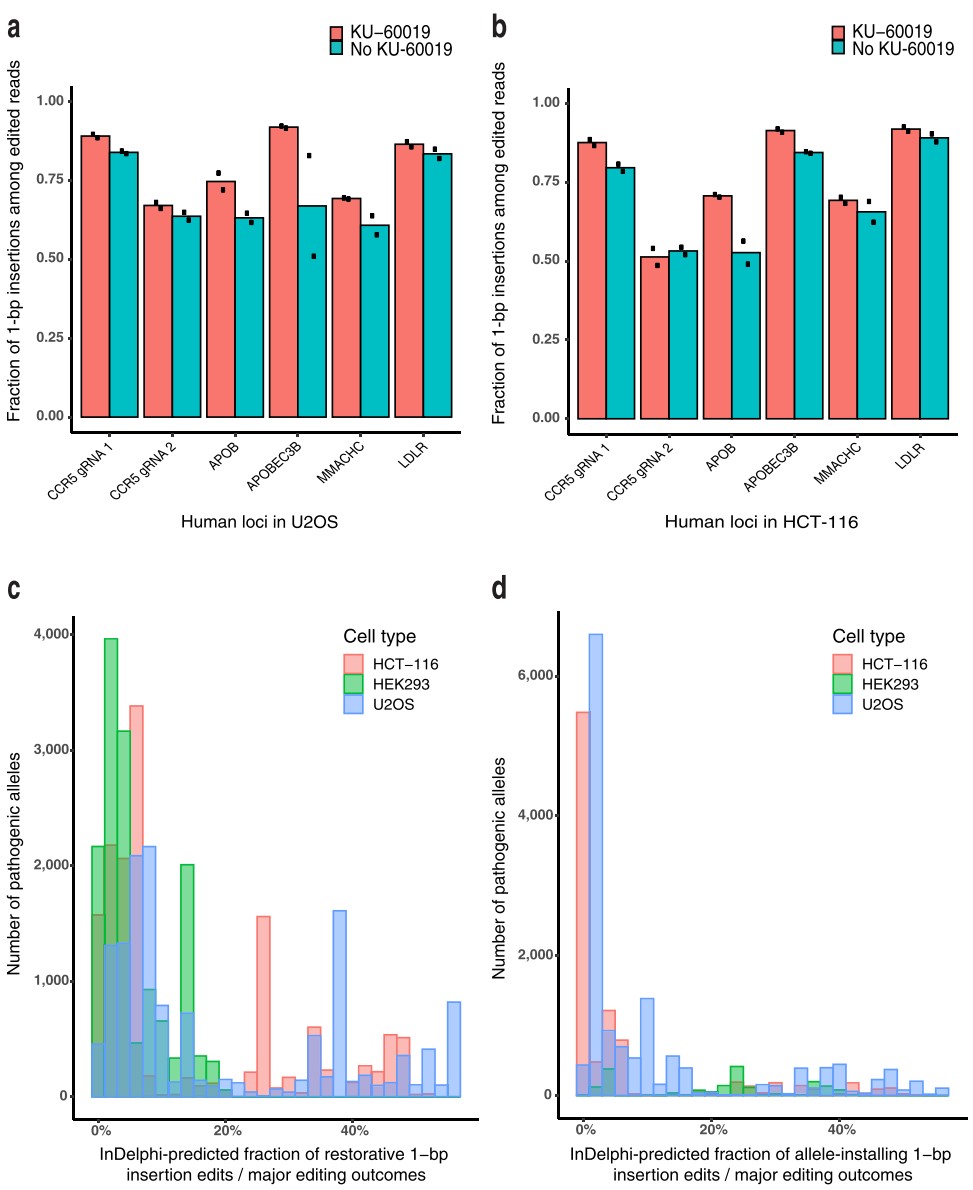

**Fig. 4 KU-60019 increases 1-bp insertions at multiple native human genomic loci, and hundreds of pathogenic alleles can be created or repaired using 1-bp insertions.** Fraction of SpCas9-NG 1-bp insertion outcomes among all edited reads at noted Precision$_{50}$ genomic loci in U2OS (**a**) and HCT-116 (**b**), $N = 2$ biologically independent samples. Histogram of inDelphi-predicted fraction of major cutsite-proximal SpCas9 editing outcomes that should lead to **c** repair to wild-type genotype for pathogenic 1-bp deletion variants from the ClinVar and HGMD databases; **d** installation of pathogenic 1-bp insertion variants from the ClinVar and HGMD databases using inDelphi models trained on cell types as listed in legend.

SaCas9, we do not know if these 11 sites fall within the typical range of 1-bp insertion frequencies for KKH-SaCas9.

We next asked whether KU-60019 could increase 1-bp insertion frequency at native Precision50 genomic sites predicted to be amenable to precise 1-bp insertion editing. We performed SpCas9-NG[29] editing at six such native Precision50 genomic loci in HCT-116 and U2OS cells in the presence or absence of KU-60019. The fraction of 1-bp insertion outcomes among all edited outcomes was increased at all six loci in each cell type, and 1-bp insertions among all edited outcomes exceeded 80% in three of six loci in HCT-116 and three of six loci in U2OS. There was a significant increase in average 1-bp insertion frequency in these six human loci in U2OS (Fig. 4a).

Finally, in order to assess the scope of 1-bp insertion editing in context of human disease-associated variants, we performed an analysis of pathogenic 1-bp deletions from the ClinVar and HGMD databases of human disease-associated variation, using inDelphi to predict the frequency with which each variant can be corrected to wild-type genotype through template-free SpCas9 editing[12,30,31]. Of the >8000 reported pathogenic 1-bp deletion mutations, the inDelphi U2OS-trained model predicts that 709 of these alleles can be repaired to wild-type with a 1-bp insertion mutation in over 50% of all edited cutsite-proximal products (Fig. 4c, Supplementary Data File 7). We performed further analysis to predict the frequency of introducing ClinVar/HGMD pathogenic 1-bp insertions using SpCas9. We found that 514 of >4000 pathogenic 1-bp insertion mutations are predicted to be created in over 50% of all edited products (Fig. 4d, Supplementary Data File 8). inDelphi models trained in HCT-116 and HEK293 cells achieved similar results (Fig. 4c, d). These analyses highlight

the importance of approaches that can increase the frequency of CRISPR-Cas9-induced 1-bp insertions.

## Discussion

In this work, we show that the ATM inhibitor KU-60019 reproducibly skews the mutational outcomes of both SpCas9-NG and KKH-SaCas9 toward 1-bp insertions across dozens of native and synthetic sites in three human and mouse cell lines. KU-60019 is a small molecule inhibitor of ATM kinase, and we have shown that three other small molecule ATM inhibitors as well as *Atm* gene knockout bias toward 1-bp insertion repair outcomes. ATM has been shown to play a role in recruiting proteins, such as NBS1 to repair DSBs, as well as stimulating checkpoint inhibitors, including *TP53* and *CHK2*[32,33]. Previous work has found that the inhibition of ATM decreases the frequency of DSB repair[34,35], in line with our finding that ATM inhibition decreases the fraction of edited alleles. While these findings demonstrate the centrality of ATM in cellular DNA repair processes, they do not clearly illuminate why KU-60019 specifically increases the frequency of 1-bp insertion outcomes. Thus, this work brings to light a role for ATM in skewing CRISPR-Cas9 outcomes toward 1-bp insertions. It will be interesting to determine whether ATM also plays a role in determining mutational outcomes of naturally occurring DSBs as well as to identify which of the many downstream targets of ATM are important in this role.

The ability to increase CRISPR-Cas9 1-bp insertion outcomes promises to improve precision genome editing applications. For instance, biasing outcomes toward 1-bp insertions will enable reliably increased CRISPR-Cas9 knockout efficiency given that MH deletions are more likely to retain frame. In addition, through analysis of the ClinVar and HGMD datasets, we show that there is a strong need for methods of increasing 1-bp insertion repair outcomes in order to enable increased precision of installing and correcting pathogenic variants. In this study, we demonstrate that KU-60019 reproducibly increases the relative frequencies of 1-bp insertion-mediated repair outcomes for 19 pathogenic human 1-bp deletion alleles.

While the possible value of CRISPR-Cas9 outcome-biasing molecules is clear, this study identifies some limitations to the use of KU-60019 and ATM inhibition toward this end. For instance, doses of KU-60019 required to alter 1-bp insertion outcome frequency dose-dependently decrease the editing efficiency of Cas9. This trade-off between product purity and editing efficiency, while it allows for more precise outcome distributions, precludes the use of KU-60019 to generate homogeneous populations of cells with 1-bp insertions. We observed decreased editing efficiency after *Atm* knockout and with three additional small molecule ATM inhibitors, suggesting that ATM function positively contributes to mutagenic CRISPR-Cas9 repair outcomes, predominantly MH deletion and non-MH deletion outcomes. All other existing CRISPR-Cas9 outcome-biasing small molecules rely on blocking key components of cellular DNA repair, such as DNA-PK, ATM, and TP53[36,37], and our assessment of 38 small molecules associated with DNA repair found that many of these compounds decrease Cas9 editing frequency. Moreover, the benefits to product purity associated with utilizing such molecules in therapeutic Cas9 editing paradigms would almost certainly be outweighed by the potential oncogenic risks[38]. Despite these limitations, the ability to increase Cas9-dependent 1-bp insertions through KU-60019-mediated inhibition of ATM adds a potent new tool to the genome editing toolbox.

## Methods

**Small molecule screening**. A small molecule library containing 487 compounds (SelleckChem, custom-made, compound information in Supplementary Data File 2) was resuspended at 10 mM in DMSO. mESC cells with stable integration of

LDLR1662-1669dupGCTGGTGA-P2A-GFP (LDLR-Dup, Supplementary Data File 1) have been previously described[1] and were cultured as described previously. For library screening, LDLR-Dup mESCs were plated at $6.67 \times 10^4$ cells per well on a gelatin-coated 96-well plate. Cells were transfected with a vector containing Cas9, a LDLR-Dup-targeting gRNA, and Neomycin resistance (p2T CAG Cas9 h7SK sgLDLR264 NeoR 5'-GACATCTACTCGCTGGTGAGC-3') using Lipofectamine 3000 (Thermo Fisher) under standard conditions in the presence of either 2 or 8 µM of a unique small molecule from the screen. Each compound was tested in triplicate at each concentration, and each plate contained eight or more wells with DMSO alone as controls. After 24 h of transfection, the media were supplemented with G418 (Thermo Fisher, used at 250 µg/mL) and either 2 or 8 µM of a unique small molecule from the screen. Starting 72 h after transfection, normal 2i mESC media were replaced in wells daily. After 7 days of transfection, LDL DyLight 550 was added at 1:500 to cells in OptiMEM (Thermo Fisher) and incubated for 3–6 h. Following a PBS wash, 0.25% trypsin was added to detach cells from their respective wells; the cells were then quenched with DMEM + 10% FBS + 2 mM EDTA + DAPI (0.5 µg/mL). Cells were transferred to 96-well V-bottom plates and run on a BD FACSymphony using 96-well plate mode, with over 10,000 cells collected per condition. Flow cytometry data were analyzed using either BD FACSDiva, FlowJo, or FCS Express software to determine the fraction of GFP+ cells. To determine significant changes of GFP+ population in small molecule vs. DMSO-treated samples, we compared GFP+ populations across three replicates, using Welch's *t*-test ($p < 0.05$). We calculated $\log_2$-fold change as the percent GFP+ population in the small molecule treatment relative to percent GFP+ in the DMSO treatment (data shown in Supplementary Data File 3).

**48-site SpCas9 gRNA:target library testing**. mESC and U2OS cells containing the 48-site library have been described[20]. Cells were then transfected using Lipofectamine 3000 with p2T CAG SpCas9 BlastR (Addgene 107190) plasmid and one of the 38 small molecules (concentrations and details listed in Supplementary Data File 4) or DMSO as a control. For U2OS, media were replaced after 6 h with media containing the small molecule. After 24 h of transfection, media were replaced with fresh media +10 µg/mL Blasticidin (mESC) or 2.5 µg/mL (U2OS) and the small molecule. After 72 h of transfection, the media containing the small molecules were removed and replaced with regular media. After 5 days of transfection, cells were harvested for gDNA using Purelink Genomic DNA mini kit (Thermo Fisher). Library prep was performed as described[20]. Illumina Nextseq was performed with >23-nt read 1 and >60-nt read 2, collecting >1 million reads per sample.

**Drug toxicity analysis**. To assess the toxicity of the small molecules, mESC cells with stably integrated gRNAs from the 48-site library were transfected with p2T CAG Cas9 BlastR and KU-0060648, KU-60019, or DMSO as indicated. After 24 h of transfection, cells were trypsinized, stained with 0.5 µg/mL propidium iodide (PI, Biolegend), and subjected to flow cytometry using a Cytek DXP11 using a consistent flow-rate. Data were analyzed using BD FACSDiva and FCS Express software, and the PI gating parameters were obtained using a live-cell gate set using untreated cells and 0.25% Triton X-100-treated cells (Supplementary 4b).

**KKH-SaCas9 12-site library testing**. mESCs containing the SaCas9 12-site library have been described previously[1] (Supplementary Data File 9). Cells were transfected with p2T CAG KKH-SaCas9 BlastR (Addgene 107189) in the presence of 2 µM KU-60019 or DMSO in a six-well plate. After 24 h, the transfection media were removed and selection was started with 2i media + 10 µg/mL Blasticidin plus small molecule. After 72 h, all drugs were removed, and cells were harvested for genomic DNA after 7 days using Purelink Genomic DNA mini kit (Thermo Fisher). Library prep was performed according to previous protocols[1], and utilized a unique primer set for each locus (Supplementary Data File 10). Illumina Nextseq was performed with >23-nt read 1 and >60-nt read 2, collecting >1 million reads per sample.

**Targeting native human loci**. U2OS and HCT-116 cells stably expressing SpCas9-NG were plated in 12-well format and transfected with previously published gRNAs targeting five human genomic loci (coding sequences in CCR5, APOB, APOBEC3B, MMACHC, LDLR, Supplementary Data File 11), each in two biological replicates in the presence of KU-60019 or DMSO. Transfection media were removed after either 6 h (U2OS) or 24 h (HCT-116), followed by addition of hygromycin at 33 or 75 µg/ml for U2OS and HCT-116, respectively. HCT-116 were split after 24 h of Hygromycin treatment to aid selection. KU-60019 was removed and replaced with regular media 72 h after transfection, and cells were harvested for genomic DNA 5 days after transfection.

**mESC Atm knockout testing**. mESC containing an integrated CAG mCherry-P2A-GFP-PuroR cassette were plated onto six wells, and transfected with gRNA plasmids (sg*Atm*: GTAAGTCATATAGGAAGCCGA or sgControl: GTAGCCCA GGTGTGCAGGCT) cloned into p2T U6 sg2xBbsI HygroR (Addgene #71485), as well as p2T CAG Cas9-MA BlastR (SpCas9 plasmid, Addgene #107190) using the standard Lipofectamine 3000 protocol. After 24 h of transfection, the media were exchanged for 2i with 0.1 mg/mL hygro + 10 µg/mL Blasticidin. Media with selection drugs were exchanged daily until 72 h after transfection, and cells were

split as needed. At 72 h, cells from each six well were split to multiple 12-wells with normal 2i media, and subsequently transfected with one of the three GFP-targeting p2T U6 sg2xBbsI HygroR gRNA plasmids (sgGFP1: GCCCATCCTGGTCGA GCTGGA, sgGFP2: GATCCGCCACAACATCGAGGA, and sgGFP3: GCTGAAGCACTGCACGCCGT), as well as p2T CAG Cas9-MA BlastR plasmid again. After 24 h of this subsequent transfection, media were replaced with 2i media + hygro + Blasticidin, and selection was continued until 72 h. After 4–7 days, when wells were sufficiently confluent (>20%), gDNA was collected using a Purelink Genomic DNA mini kit. $N = 2$ for each of the six conditions.

**mESC 48-site additional ATM-inhibitors testing**. mESCs were cultured in the presence of varying concentrations for each of the three ATM-inhibitors (CP-466722, AZD1390, and AZ32) for 5 days in order to determine the maximum concentration at which cell growth was not compromised for each drug; we observed a maximum tolerated concentration of 1 μM for AZD1390, 2 μM for AZ32, and 1 μM for CP-46722. mESCs with the 48-site library were plated in triplicates for each condition within gelatin-coated six-well plates. The following day, each well was transfected with p2T CAG Cas9-MA blastR plasmid using standard Lipofectamine 3000 protocol in the presence of the respective ATM inhibitor or DMSO control. After 24 h of transfection, media were replaced with 2i media + 10 μg/mL Blasticidin + inhibitor/DMSO. After 72 h of transfection, media were replaced with regular 2i media. After 4–6 days of transfection (depending on when the wells were sufficiently confluent), gDNA was harvested using a Purelink gDNA mini kit.

**Ribonucleoprotein (RNP) SpCas9 editing**. RNP editing was performed according to manufacturer's protocol using mCherry-targeting crRNA (GGAGCCGTA-CATGAACTGAG), AltR tracrRNA (IDT), and Alt-R SpCas9 (IDT), Cas9 Plus reagent (Thermofisher), and CRISPRMAX transfection reagent. mESCs containing the p2T integrated CAG mCherry-P2A-GFP-PuroR cassette were dosed with RNP transfection mix in the presence of 2 μM KU-60019 or DMSO ($N = 3$ for each condition). After 24 h, media were changed with regular 2i media, and cells were split at 48 h to allow for further growth. At 72 h, gDNA isolation was collected using a Purelink Genomic DNA mini kit.

**ClinVar and HGMD data analysis**. Disease-associated variants were selected from the NCBI ClinVar database (downloaded September 9, 2017)[30] and the Human Gene Mutation Database (publicly available variant data before 2014.3)[31] for computational screening using the inDelphi models trained on U2OS, HCT-116, and HEK293 cells[12]. ClinVar variants were included where at least one submitting lab designated the clinical significance as 'pathogenic' or 'likely pathogenic' and no submitting lab had designated the variant as 'benign' or 'likely benign'. HGMD variants were included with any disease association with the HGMD classification of 'DM' or disease-causing mutation. SpCas9 gRNAs and their cleavage sites were enumerated for each disease allele. Genotype frequency was predicted for each tuple of disease variant and unique cleavage site (Supplementary Data Files 7 and 8). For each unique variant, the single best gRNA was identified as the gRNA inducing the highest predicted frequency of repair to wild-type genotype (for 1-bp deletion variants) or allele installation (for 1-bp insertion variants).

**Nextgen sequencing analysis**. Analysis was performed on raw reads from Illumina deep sequencing. Sequencing reads were filtered to remove low-quality (Illumina average quality <28) or unmapped reads. For each sequencing read representing a CRISPR-Cas9 cutting event, the cutting genotype was identified and categorized as an insertion or deletion event; overall fractions of insertions and deletions of all lengths were then computed from the two replicates. This was carried out with a published analysis pipeline described in Shen et al.[12]. For indel and 1-bp insertion detection at native human genomic loci, Crispresso2 was used to carry out the alignment of NGS reads to the wild-type amplicon sequence, using default parameters and a Needleman Wunsch gap extended score of 0 for optimal alignment with visual inspection[39]. Reads were determined to be modified or unmodified if a 1-bp insertion was located within 2 bp from each sgRNA cleavage (default quantification window for Cas9 Crispresso2).

**General statistics and reproducability**. For comparison of two independent groups, two-sided two-sample t-tests were used for normally distributed data with equal or similar variance (Student's t-test) or unequal and dissimilar variance (Welch's t-test). A value of $p < 0.05$ was used to determine significance. For multiple comparison tests, One-way ANOVA with Tukey HSD post hoc correction was performed. All experiments were conducted according to the provided methods only once, and all data collected is reflected within the supplementary data files.

**Reporting summary**. Further information on research design is available in the Nature Research Reporting Summary linked to this article.

## Data availability

All high-throughput sequencing data have been deposited in the NCBI Sequence Read Archive database under accession codes PRJNA744770 and PRJNA658607. Processed data have been deposited under the following DOIs: https://doi.org/10.6084/m9.figshare.12844577, https://doi.org/10.6084/m9.figshare.12844625, https://doi.org/10.6084/m9.figshare.12844631, https://doi.org/10.6084/m9.figshare.14046581.

In addition, this study accessed and utilized data from the ClinVar Database and Human Genome Mutation Database.

## Code availability

All data processing, analysis, and modeling codes are available at www.github.com/gifford-lab/inDelphi-dataprocessinganalysis. The inDelphi model is available online at https://indelphi.giffordlab.mit.edu/.

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

## Acknowledgements

The authors would like to thank the Harvard Medical School Biopolymers Facility for sequencing, and the Harvard Medical School Department of Immunology Flow Cytometry Facility for flow cytometry related technical assistance. The authors acknowledge funding from 1R01HG008754 (R.I.S.), 1R21HG010391 (R.I.S.), NWO (R.I.S.), American Cancer Society (R.I.S.), American Heart Association (R.I.S.), and Qatar Biomedical Research Institute (R.I.S.).

## Author contributions

Conceptualization, methodology, writing—original draft and writing—reviewing and editing: H.C.B.C., S.B., R.I.S. and D.K.G.; Investigation and validation: H.C.B.C., S.C., S.B., R.I.S. and S.Z.; Software, formal analysis, and visualization: H.C.B.C., S.B., B.H. and M.W.S.; Funding acquisition and supervision: R.I.S. and D.K.G.

## Competing interests

The authors declare no competing interests.
