## [Peer Review File · Nature Communications]

Reviewers' Comments:

Reviewer #1:

Remarks to the Author:

Precise gene editing is essential for therapeutic applications and many other important applications of gene editing but is often inefficient. Together with other labs, the authors previously found that the pattern of mutations induced by DNA cleavage with CRISPR/Cas9 is highly dependent on the DNA sequence flanking the cleavage site and discovered that a significant fraction of guide RNAs lead to a predominant mutation, which is generally a 1-bp insertion or a "MH" deletion (i.e. a small deletion between microhomologies on either side of the DNA break). They showed that when such guide RNAs induce mutations of interest, they could thus be used to achieve template-free precise gene editing.

Bermudez-Cabrera et al have now performed a small screen to identify compounds that impact the pattern of mutations induced by DNA cleavage with CRISPR/Cas9 systems in order to further increase the efficiency of the so called "precision" guides. It was previously shown that inhibitors of DNA repair by NHEJ increase the proportion of MH deletions (deletions flanked by Micro-Homologies). Bermudez-Cabrera et al now report that the ATM kinase inhibitor Ku-60019 increases the proportion of 1-bp insertions while decreasing that of MH deletions.

This is an important topic and a potentially significant contribution. However, the manuscript would need to be improved and a lot of additional information should be given on the data as detailed below (listed in minor comments).

Several major issues need to be addressed:

1. As stated by the authors at the end of the discussion, although the mutation patterns are changed by the ATM inhibitor KU-60019 and the DNA-PKc inhibitor KU-0060648, the potential benefit of such modulations are compromised by a parallel reduction in the number of mutant sequences. The authors have excluded cell death as a potential explanation when testing the 48-site library. The reduction in editing efficiency, however, is a great limitation to the use of the different compounds and it would need to be further investigated. One trivial possibility is that there is a reduction of Cas9 or guide RNA expression. Might using Cas9 RNPs overcome this limitation?
2. The compounds were first tested in a reporter cell line and next on a series of 48 exogenous sites which were previously introduced in mESCs and U2OS cells. The 48-site library design is not fully documented and the manuscript submitted with this info should be provided. The sites in the library were selected because they preferentially give 1-bp insertions or MH deletions or give a mixed pattern (with 16 guides of each type in the 48-site library). Unexpectedly, the changes in the mutation patterns are not reported by type of target site but rather after aggregating the data for all 48 sites (Figures 1 and 2). In order to more finely evaluate the impact of the compounds, the primary data, including total editing efficiency, should also be given for all 48 sites as well as aggregated and analyzed for each of the 3 types of sites.
3. The editing efficiency in experiments reported here appears quite low (less than 9 % edited reads according to Figure 2). Is this because the reporter cell lines carry a high copy number of the 48-site library? In any case, it is important to test how the compounds impact mutation patterns when editing efficiency is much higher. This is relevant to practical applications of the template-free approach proposed by the authors.
4. When showing results on native sites (Figure 4), the authors should also report total editing efficiencies and document the change in editing efficiency at native sites when using KU-60019 with KKH-SaCas9 or Cas9-NG. Clearly, this information is needed to evaluate the benefit of using KU-60019. The discussion section suggests that the decrease of editing efficiency reported for the 48-site library was also observed for native sites but the information appears to be missing.
5. The possibility that ATM negatively regulates the frequency of 1-bp insertions is intriguing. It would be important to confirm that ATM is indeed involved by using ATM inhibitors other than KU-60019 and by examining the effect of inhibiting ATM gene expression. Further mechanistic insight would also significantly strengthen the manuscript.

Finally, it would be important to discuss the potential of making 1-bp insertions for correcting

pathogenic mutations or modeling genetic diseases. A genome-wide analysis of naturally occurring mutations, such as performed for MH deletions by Grajacarek et al, Nat Comm 2018, would be useful.

Minor comments

1. The reference given for the reporter construct, a key component of the screen, is not correct. Since the correct reference (Shen et al, Nature, 2018) describes several LDLR-2A-GFP reporters, the sequence of the reporter and guide RNA used here should be provided in the manuscript. How many copies of the reporter construct are there in the cell line used for the screen?
2. The claim made in the abstract that "KU-60019 increases the precision of 1-bp insertions past 80% at several native human genomic loci" does not clearly describe the contribution of Ku-60019 to this high efficiency. Please clearly make this statement in the results section and document the proportion of targets tested for which this is observed in the abstract.
3. To the best of my knowledge, the mutation pattern of SaCas9 has not been investigated as extensively as that of SpCas9. The data shown here suggest that frequencies of 1-bp insertions are much lower than for SpCas9 which would limit the relevance of SaCas9 for the method of gene editing proposed by the authors.
4. I assume that Cas9-NG needed to be used in Figure 3b because the 1-bp deletions selected cannot be targeted with standard Cas9. This should be made explicit. In fact, the text in the results section related to Figure 3b does not mention the use of Cas9-NG at all and the reference to the article describing Cas9-NG is not included nor is the lab cited in the acknowledgments section.

Reviewer #2:

Remarks to the Author:

The paper submitted by Bermudez-Cabrera and colleagues, "A small molecule ATM kinase inhibitor increases CRISPR/Cas9 1-bp insertion frequency" is of great interest to the field of CRISPR-mediated gene editing. The authors initiated this study to find small molecules capable of increasing 1-bp insertion mutations during CRISPR editing, which is thought to be a more therapeutically relevant mutation compared to either microhomology or non-microhomology mediated mutations. Overall, I found that the novel findings of the study, usefulness to the field of gene editing, and robustness warrants its publication. I do however have some comments that should be addressed prior to acceptance.

For instance, while we see from the results that there are 1bp insertions, we do not know if there are significant local rearrangements (the authors chose not to assess this in the study). As such, they should either not showcase these results as therapeutically relevant at the present time, or the experiment should be conducted with PacBio sequencing to determine if there are local rearrangements present or not.

While we can assume the ATM inhibitor indeed inhibits ATM, the authors assume that the compound's effect is only via ATM and not through some other off-target interaction. This is highlighted by the fact that there were no other ATM inhibitors that were identified in the screen. To address this, I suggest that the discussion should be edited to consider that the compound may be inhibiting other targets, leading to 1bp insertions. Another option would be to assess the percentage of 1bp insertions in an ATM knockout cell line, or using other ATM inhibitors in addition to KU-60019 to support the claim that the 1bp insertions occur via loss of function of ATM. An Excel sheet in the supplementary data with all of the compounds tested would be useful. I assume that the 50 or so highlighted in the supplementary info are the ones of interest, however it would in addition be useful to see all of those tested. This would be especially helpful for reference in figure 1a (could details be added to see where each drug tested falls on the graph? Or perhaps drugs in red could be highlighted?).

In the introduction, it is stated that the 38 compounds followed up on were those that exhibited a

significant change in LDLR-Dup GFP+ fraction, however actually those selected are the ones that led to an increase in this value. This should be stated more clearly.

A schematic of the experimental design in the figures would be beneficial to the understanding of the work.

The figures are often stretched and the labelling used is unclear (e.g. fold change, fraction of KKH-SaCas9 genotype among edited reads).

Below, we have attached a detailed point-by-point response to the reviewers' comments. We have carefully addressed every reviewer comment through additional analyses, experiments, and clarifications to the text; we believe that these changes have improved the quality of our manuscript and further strengthened our findings. To summarize the major changes to the manuscript, we have:

- Found that three additional ATM inhibitors and *Atm* knockout lead to increased 1-bp insertion frequency, strengthening our evidence that ATM pathway inhibition is responsible for this biased CRISPR-Cas9 outcome distribution.
- Found that KU-60019 increases 1-bp insertion frequency and decreases overall editing frequency when Cas9 and gRNA are delivered through a ribonucleoprotein (RNP) complex, suggesting that KU-60019's effects are unlikely to relate to manipulating expression of plasmid Cas9 or gRNA.
- Performed an analysis of pathogenic 1-bp deletion and 1-bp insertion variants using inDelphi, finding that thousands of such alleles are predicted to be installable with appreciable frequency, and there are >700 pathogenic 1-bp deletions that we predict can be corrected in >50% of cutsite-proximal edited products through template-free SpCas9 editing.
- Updated our figures to provide more intuitive legends, higher-resolution panels, and more complete information on experiments.

In the section below, our responses are listed in blue.

Reviewer 1:

Precise gene editing is essential for therapeutic applications and many other important applications of gene editing but is often inefficient. Together with other labs, the authors previously found that the pattern of mutations induced by DNA cleavage with CRISPR/Cas9 is highly dependent on the DNA sequence flanking the cleavage site and discovered that a significant fraction of guide RNAs lead to a predominant mutation, which is generally a 1-bp insertion or a "MH" deletion (i.e. a small deletion between micro-homologies on either side of the DNA break). They showed that when such guide RNAs induce mutations of interest, they could thus be used to achieve template-free precise gene editing.

Bermudez-Cabrera et al have now performed a small screen to identify compounds that impact the pattern of mutations induced by DNA cleavage with CRISPR/Cas9 systems in order to further increase the efficiency of the so called "precision" guides. It was previously shown that inhibitors of DNA repair by NHEJ increase the proportion of MH deletions (deletions flanked by Micro-Homologies). Bermudez-Cabrera et al now report that the ATM kinase inhibitor Ku-60019 increases the proportion of 1-bp insertions while decreasing that of MH deletions.

This is an important topic and a potentially significant contribution.

Response: We thank the reviewer for the positive feedback and their assertion of our manuscript's potential impact.

However, the manuscript would need to be improved and a lot of additional information should be given on the data as detailed below (listed in minor comments).

Several major issues need to be addressed:

1. As stated by the authors at the end of the discussion, although the mutation patterns are changed by the ATM inhibitor KU-60019 and the DNA-PKc inhibitor KU-0060648, the potential benefit of such modulations are compromised by a parallel reduction in the number of mutant sequences. The authors have excluded cell death as a potential explanation when testing the 48-site library. The reduction in editing efficiency, however, is a great limitation to the use of the different compounds and it would need to be further investigated. One trivial possibility is that there is a reduction of Cas9 or guide RNA expression. Might using Cas9 RNPs overcome this limitation?

Response: To address whether decreased gRNA or Cas9 expression leads to lower overall editing efficiency, we have addressed the effects of KU-60019 in the context of RNP-based CRISPR-Cas9 editing. Upon testing this RNP approach and analyzing the mutational distribution, we found that the presence of KU-60019 increases the relative frequency of 1-bp insertions ($p < .001$, Supplementary Fig. 5a) while decreasing the total editing efficiency (Supplementary Fig. 5b, fold change: 0.38). The similarity of our results in RNP and plasmid-based editing paradigms suggests that KU-60019 is more likely decreasing editing efficiency through impacting DNA repair pathways.

We agree that the decreased editing efficiency in the presence of ATM inhibition (which we now show using four small molecules and genetic *Atm* knockout) is an important caveat to the therapeutic utility of the increased frequency of 1-bp insertions. We have made this point clearly in our discussion, further noting that previous studies have found decreased endogenous DSB repair in the presence of ATM inhibition.

2. The compounds were first tested in a reporter cell line and next on a series of 48 exogenous sites which were previously introduced in mESCs and U2OS cells. The 48-site library design is not fully documented and the manuscript submitted with this info should be provided.

Response: We have now included this information in Supplementary Table 5, in which we report each gRNA and target sequence within the 48-site library. We also note that we have now published a manuscript that more fully describes the design and analysis of the 48-site library (<https://doi.org/10.1021/acscentsci.0c00129>) and have cited this publication when referencing the 48-site library.

2a. The sites in the library were selected because they preferentially give 1-bp insertions or MH deletions or give a mixed pattern (with 16 guides of each type in the 48-site library). Unexpectedly, the changes in the mutation patterns are not reported by type of target site but rather after aggregating the data for all 48 sites (Figures 1 and 2). In order to more finely evaluate the impact of the compounds, the primary data, including total editing efficiency, should also be given for all 48 sites as well as aggregated and analyzed for each of the 3 types of sites.

Response: We have now provided the fraction of MH deletion and 1-bp insertion outcomes for each gRNA within the 48-site library in the presence of DMSO, KU-60019, and KU-0060648 (Supplementary Table 6a-b). This analysis shows that these compounds reproducibly bias the repair outcome distribution toward MH deletion/1-bp insertion across dozens of targets in both mESCs and U2OS cells. We have also

plotted the fraction of 1-bp insertion outcomes in the presence or absence of KU-60019 for 8 pathogenic 1-bp deletion targets that are included in the 48-site library (Supplementary Fig. 4), showing that there is increased 1-bp insertion at 7 of the 8 targets.

3. The editing efficiency in experiments reported here appears quite low (less than 9 % edited reads according to Figure 2). Is this because the reporter cell lines carry a high copy number of the 48-site library? In any case, it is important to test how the compounds impact mutation patterns when editing efficiency is much higher. This is relevant to practical applications of the template-free approach proposed by the authors.

Response: We do not know why the editing efficiency in the 48-site library tends to be low-- we have found that library approaches such as this do tend to produce lower editing efficiency than individual targeting experiments, possibly because of limiting amounts of gRNA transcript. However, our data suggests that KU-60019 increases the fraction of 1-bp insertion outcomes in paradigms of higher editing efficiency as well: half of the native pathogenic sites targeted in our U2OS KKH-SaCas9 loci screen yielded >20% editing efficiency (maximum 60% editing efficiency), and KU-60019 treatment robustly increased the frequency of 1-bp insertions for all of these sites (Fig. 4a, Supplementary Fig. 6c).

4. When showing results on native sites (Figure 3), the authors should also report total editing efficiencies and document the change in editing efficiency at native sites when using KU-60019 with KKH-SaCas9 or Cas9-NG. Clearly, this information is needed to evaluate the benefit of using KU-60019. The discussion section suggests that the decrease of editing efficiency reported for the 48-site library was also observed for native sites but the information appears to be missing.

Response: We have added an additional figure, Supplementary Figure 6, to show the editing efficiency at each native locus treated with SpCas9 and for the KKH-SaCas9 library. We have also made sure to include edited fractions for each editing experiment described in the manuscript.

5. The possibility that ATM negatively regulates the frequency of 1-bp insertions is intriguing. It would be important to confirm that ATM is indeed involved by using ATM inhibitors other than KU-60019 and by examining the effect of inhibiting ATM gene expression. Further mechanistic insight would also significantly strengthen the manuscript.

Response: This is indeed an important point. In order to address whether KU-60019 is biasing 1-bp insertion mutations through ATM inhibition, we compared the mutational distribution in mESC with our 48-site integrated library in the presence of either KU-60019 or three other ATM inhibitors, showing that all three other small molecule ATM inhibitors reproducibly bias DSB repair outcomes towards 1-bp insertions. We have also performed Cas9 targeting in mESCs previously targeted with Cas9-induced *Atm* knockout, and find that *Atm* knockout also reproducibly biases DSB repair outcomes toward 1-bp insertions. These two additional experiments strongly suggest that ATM inhibition through either small molecule or genetic means is sufficient to bias repair outcomes toward 1-bp insertions. We hypothesize that promotion of 1-bp insertion by KU-60019 is thus likely to occur through its known ability to inhibit ATM kinase.

6. Finally, it would be important to discuss the potential of making 1-bp insertions for correcting pathogenic mutations or modeling genetic diseases. A genome-wide analysis of naturally occurring mutations, such as performed for MH deletions by Grajacarek et al, Nat Comm 2018, would be useful.

Response: In order to demonstrate the clinical relevance of 1-bp insertions, we have performed an analysis of pathogenic 1-bp deletions from the ClinVar and HGMD databases along with the inDelphi-predicted frequency of correction through SpCas9-nuclease editing. We have provided a similar analysis of pathogenic 1-bp insertions that we predict can be installed using SpCas9-nuclease editing. We provide a summary of these results in Figure 4 and provide the data as Supplementary Tables 7 and 8. Overall, there are thousands of reported pathogenic 1-bp deletion mutations, and inDelphi predicts that 709 such alleles can be repaired by 1-bp insertion in over 50% of cutsite-proximal edited products using the U2OS model. A similar number (514) of 1-bp insertion alleles are predicted to be installable in over 50% of edited products. This analysis highlights the importance of approaches to increase the frequency of 1-bp insertion editing such as KU-60019 and other approaches stemming from our findings.

Minor comments

1. The reference given for the reporter construct, a key component of the screen, is not correct. Since the correct reference (Shen et al, Nature, 2018) describes several LDLR-2A-GFP reporters, the sequence of the reporter and guide RNA used here should be provided in the manuscript. How many copies of the reporter construct are there in the cell line used for the screen?

Response: We have now provided this information in Supplementary Table 1. Cells should have a single copy of the reporter construct, as we performed limiting dilution transposition as described in the methods which we have shown to integrate a single copy of the reporter per cell.

2. The claim made in the abstract that “KU-60019 increases the precision of 1-bp insertions past 80% at several native human genomic loci” does not clearly describe the contribution of Ku-60019 to this high efficiency. Please clearly make this statement in the results section and document the proportion of targets tested for which this is observed in the abstract.

Response: We have revised this sentence in the abstract to read “Notably, KU-60019 increases the relative frequency of 1-bp insertions to over 80% of edited alleles at several native human genomic loci”. We believe this to be an accurate representation of data presented in Figure 4.

3. To the best of my knowledge, the mutation pattern of SaCas9 has not been investigated as extensively as that of SpCas9. The data shown here suggest that frequencies of 1-bp insertions are much lower than for SpCas9 which would limit the relevance of SaCas9 for the method of gene editing proposed by the authors.

Response: It is true that mutational outcomes of SaCas9 have not been investigated as extensively. It does not follow that SaCas9 has a lower propensity to induce 1-bp insertions. Because we do not have the benefit of a mutational outcome prediction model like inDelphi when choosing which loci to investigate with SaCas9, we do not know if we chose sites with lower than average 1-bp insertion frequency. We can conclude from these experiments that KU-60019 increases 1-bp insertion outcomes

induced by SaCas9 but refrain from concluding anything regarding the fraction of 1-bp insertions provided by SaCas9 in general. We have updated the text to explain this more clearly.

4. I assume that Cas9-NG needed to be used in Figure 3B because the 1-bp deletions selected cannot be targeted with standard Cas9. This should be made explicit. In fact, the text in the results section related to Figure 3B does not mention the use of Cas9-NG at all and the reference to the article describing Cas9-NG is not included nor is the lab cited in the acknowledgments section.

Response: for the 48-site library and native screens, we opted to use SpCas9-NG instead of SpCas9 since we had cultures of mESC, HCT-116, and U2OS cell lines that had previously been transfected, selected, and FACS sorted for that specific Cas9 variant. We have updated our results section, figure titles, and figure panels to clearly distinguish SpCas9-NG experiments, and have added a reference to the original paper describing SpCas9-NG.

Reviewer #2:

The paper submitted by Bermudez-Cabrera and colleagues, “A small molecule ATM kinase inhibitor increases CRISPR/Cas9 1-bp insertion frequency” is of great interest to the field of CRISPR-mediated gene editing. The authors initiated this study to find small molecules capable of increasing 1-bp insertion mutations during CRISPR editing, which is thought to be a more therapeutically relevant mutation compared to either microhomology or non-microhomology mediated mutations. Overall, I found that the novel findings of the study, usefulness to the field of gene editing, and robustness warrants its publication.

Response: We thank the reviewer for the positive feedback.

I do however have some comments that should be addressed prior to acceptance.

*For instance, while we see from the results that there are 1bp insertions, we do not know if there are significant local rearrangements (the authors chose not to assess this in the study). As such, they should either not showcase these results as therapeutically relevant at the present time, or the experiment should be conducted with PacBio sequencing to determine if there are local rearrangements present or not.

Response: We agree that without considering rearrangements, we should recommend caution in the interpretation of our results. We have expanded our use of the term cutsite-proximal editing outcomes to clarify what our approaches are capable of measuring. We have attempted to further caveat all mentions of therapeutic relevance of utilizing KU-60019 in context of Cas9 as a result. We agree that examining the contribution of ATM inhibition on rearrangements is an important future step, and we have added text to this effect.

*While we can assume the ATM inhibitor indeed inhibits ATM, the authors assume that the compound's effect is only via ATM and not through some other off-target interaction. This is highlighted by the fact that there were no other ATM inhibitors that were identified in the screen. To address this, I suggest that the discussion should be edited to consider that the compound may be inhibiting other targets,

leading to 1bp insertions. Another option would be to assess the percentage of 1bp insertions in an ATM knockout cell line, or using other ATM inhibitors in addition to KU-60019 to support the claim that the 1bp insertions occur via loss of function of ATM.

Response. As mentioned in our response to reviewer 1, we have performed experiments to address this important point. In order to address whether KU-60019 is biasing 1bp-insertion mutations through ATM inhibition, we compared the mutational distribution in mESC with our 48-site integrated library in the presence of either KU-60019 or three other ATM inhibitors, showing that all three other small molecule ATM inhibitors reproducibly bias DSB repair outcomes towards 1-bp insertions. We have also performed Cas9 targeting in mESCs previously targeted with Cas9-induced *Atm* knockout, and find that *Atm* knockout also reproducibly biases DSB repair outcomes toward 1-bp insertions. These two additional experiments strongly suggest that ATM inhibition through either small molecule or genetic means is sufficient to bias repair outcomes toward 1-bp insertions. We hypothesize that promotion of 1-bp insertion by KU-60019 is thus likely to occur through its known ability to inhibit ATM kinase.

*An Excel sheet in the supplementary data with all of the compounds tested would be useful. I assume that the 50 or so highlighted in the supplementary info are the ones of interest, however it would in addition be useful to see all of those tested. This would be especially helpful for reference in Figure 1A (could details be added to see where each drug tested falls on the graph? Or perhaps drugs in red could be highlighted?).

Response: We have added the results of all small molecules tested in Figure 1a as Supplementary Table 3.

*In the introduction, it is stated that the 38 compounds followed up on were those that exhibited a significant change in LDLR-Dup GFP+ fraction, however actually those selected are the ones that led to an increase in this value. This should be stated more clearly.

Response: We followed up on compounds that had a significant change in either direction. As an example, KU-60019 decreased MH deletion repair, which we show is a function of increasing 1-bp insertion outcomes. Other compounds presumably decreased MH deletion repair through decreasing overall editing frequency, as explained in the text.

*A schematic of the experimental design in the figures would be beneficial to the understanding of the work.

Response: A schematic of the reporter construct and screen design is now included as Supplemental Figure 1.

*The figures are often stretched and the labelling used is unclear (e.g. fold change, fraction of KKH-SaCas9 genotype among edited reads).

Response: We have properly embedded the figures now in order to prevent the figure-stretching observed in our initial submission, and we have revised our labeling and figure legends to more clearly describe our results.

Reviewers' Comments:

Reviewer #1:

Remarks to the Author:

The manuscript by Bermudez-Cabrera et al represents an important contribution to the field of precision genome editing. The revised manuscript includes additional experiments, clarifications and access to primary data as requested and I recommend its publication.

Reviewer #2:

Remarks to the Author:

The authors have addressed all of the points raised and have gone beyond what was expected (e.g., ATM knockout and 3 addition ATM inhibitors). I congratulate the authors on an excellent study.